# OpenReview forum: "Probing the robustness of large language models safety to latent perturbations"
_ICLR.cc/2026/Conference — ICLR 2026 Conference Withdrawn Submission_

### Official Review · Reviewer_X6St · 2025-10-27

**Soundness:** 3
**Presentation:** 4
**Contribution:** 4
**Rating:** 8
**Confidence:** 4

**Summary:**

This paper works on identifying vulnerabilities to simple latent space perturbations in LLMs. It proposes and validates a hypothesis connecting three things: the success of NLL probes, a model's vulnerability to latent-space perturbations, and the model's vulnerability to harmful behavior extraction attacks. Finally, the paper introduces a form of adversarial training under these latent space perturbations and shows that it improves model robustness to attacks.

**Strengths:**

S1: I think that NLL probing is clever, and I am glad that the authors got it to work as a useful proxy for the latent attackability of latent activations.

S2: I like the experiments to combine ASA with other attacks.

S3: Overall, I think that the work is clever and working on a good problem.

**Weaknesses:**

W1: I think that claim number 1 at the end of the introduction is probably an over claim. I think this has been demonstrated before:
- https://arxiv.org/abs/2412.09565
- https://arxiv.org/abs/2312.02780
- https://arxiv.org/abs/2502.05209
- https://arxiv.org/abs/2403.05030
- https://arxiv.org/abs/2406.11717

W2: In Figure 2, INIT correlates with both MASR and PASR. This seems very important. It validates the hypothesis that vulnerability to ASA is indicative of weak alignment overall. You should plot this and point it out. This could be framed as a core contribution of the paper. NB that https://arxiv.org/abs/2502.05209 made a similar observation.

W3: Ideally, LAPT would be compared against similar baselines like LAT (https://arxiv.org/abs/2407.15549), MixAT (https://arxiv.org/abs/2505.16947), and RFAT (https://arxiv.org/abs/2409.20089)

Minor (don't reply to these)
- Missing citations
  - https://arxiv.org/abs/2407.15549
  - https://arxiv.org/abs/2412.09565
  - https://arxiv.org/abs/2406.05946
- Small figures and fig text -- some could be made bigger or rearranged to use multiple rows
- I am somewhat confused by the differing layers in table 3

**Questions:**

See "weaknesses"

---

### Official Review · Reviewer_aW35 · 2025-10-27

**Soundness:** 2
**Presentation:** 2
**Contribution:** 2
**Rating:** 2
**Confidence:** 3

**Summary:**

* This paper tests a way to do adversarial testing and training in latent space
* The authors find that models are vulnerable to latent space attacks (e.g., when optimized via gradients)
* The authors also test a way to train against such attacks, finding that they’re able to maintain the performance of models on various tasks like GSM8k

**Strengths:**

* The general area is pretty promising (using latent space attacks to find adversarial vulnerabilities and train them away)
* The particular approach seems like a reasonable approach to have tested

**Weaknesses:**

* It seems obvious that latent adversarial attacks will work for eliciting harmful info from models (I think this has also been shown in prior work). I think it doesn’t have major implications to me as well, since models aren’t trained to be robust to latent space attacks. The existence of latent attacks doesn’t clearly imply anything about the model’s actual robustness on the real distribution of inputs (which is what matters)
* I like the motivation to train against these attacks though, rather than to use them to test robustness. This is because training against these attacks may generalize to training away input-space attacks. This would be advantageous in the case that input-space attacks are harder or more expensive to find than latent space attacks (which I could imagine being the case). However, an important baseline here is to test that training against black-box red teaming methods (like automated red teaming with LLMs) is actually less effective (either in terms of final attack success rate, or in terms of compute cost). This is a pretty key baseline which is missing from the paper, which makes the results hard to interpret
* I may have missed this in the paper but it’s unclear to me if the adversarially trained model retains it’s full capabilities across a wide range of tasks, and e.g., whether the samples look normal. I think just looking at performance degradation on a couple of tasks isn’t sufficient, and I think it would be good to look at performance and sample quality across a wide range of tasks
* I found the paper hard to follow, in terms of the specific technique, what the evaluations were, etc. This made it a bit harder to know if I understood the method, evals, and paper overall well enough to get a sense of whether it would be useful for some of the applications I had in mind for latent adversarial training

**Questions:**

1. Do you know how adv training with ASA compares to adv training with black-box red teaming approaches?
2. How much capabilities degradation does adv training result in for held-out tasks?
3. How much does sample quality degrade with adv training (e.g., when talking to the model as a chat assistant)?

---

### Official Review · Reviewer_r9Uy · 2025-11-01

**Soundness:** 3
**Presentation:** 2
**Contribution:** 2
**Rating:** 2
**Confidence:** 4

**Summary:**

This paper investigates vulnerabilities in LLM safety alignment by introducing Activation Steering Attack (ASA), which perturbs hidden activations at intermediate layers to trigger unsafe responses. The authors demonstrate that minor latent perturbations can compromise safety-aligned models, revealing fundamental weaknesses in current alignment strategies. They propose a probing method using Negative Log-Likelihood (NLL) to identify vulnerable directions, construct ASABench (a benchmark with 4,862 validated attack instances), and introduce Layer-wise Adversarial Patch Training (LAPT) as a defense mechanism. Experiments across 12 open-source models show ASA's effectiveness, with the gradient-based variant (ASAgrad) achieving particularly high attack success rates.

**Strengths:**

The core method here is sound and worth knowing about. It seems realistic that malicious actors will want to steer open-source models to do tasks their developers didn't intend, so benchmarking and defending against this seems like a necessary and timely area for intervention. The core technical approach is sound and the experimental methodology is reasonable. The NLL-based probing method provides a principled way to identify vulnerable directions in latent space, and the normalization scheme (Eq. 1) is well-motivated. The gradient-based extension (ASAgrad) follows logically from the random baseline. However, I have some concerns about the threat model's practical relevance and certain experimental choices.

The paper's main technical contributions - ASA, ASAgrad, and LAPT - are adequately supported by experiments across multiple models. The use of QwQ-32B as an automatic judge is justified through the accuracy comparison in Appendix L.

Overall this definitely hits on a vulnerability that hasn't been explored, and which seems quite hard to defend models against misuse on. It's also cool to see the composition with GCG, and that ASA can make other attacks significantly more potent. Their defense method seems both quite strong and doesn't degrade other model capabilities.

**Weaknesses:**

While it's true that I'm not aware of other work exploring this axis of attack on models, I'd like the authors to explain more of why or when this would be a realistic threat model. This degree of whitebox access isn't common in commercial AI deployments, especially among models that top the leaderboards on various relevant axes. It's also unclear why attackers with whitebox access would use activation steering over other methods to make harmless models harmful.

The methods here are also mostly incremental, since activation steering has been around for a long time. That means the paper needs to lean more on the defense results, which are a little thin if that's most of the contribution the paper has to stand on, in addition to the benchmark of attacks. There's a fair amount of experimental work going on in the paper, but I'm not sure that it's enough to justify the low-ish novelty.

It feels minor at first but on the whole quite distracting that the paper is not particularly well written. For example, there are some pretty confusing turns of phrase in this paper. One notable example is a focus on "safety alignment", which is a nonstandard term that I've not heard anywhere else. This plus many typos (notably that many verbs are incorrectly conjugated) make me think that the authors should run the paper through another pass of copyediting as a basic first step before publication. The writing also occasionally lacks precision (e.g., "shallow nature of existing alignment methods" is vague and doesn't tell me enough about what's different about this method)

**Questions:**

1. I'm struggling to understand when ASA would be a realistic attack in practice. You assume white-box access to intermediate activations, which isn't common in commercial deployments - especially for leading models. Can you walk me through a concrete scenario where an adversary would have this level of access but wouldn't just use fine-tuning attacks or directly edit weights? What makes activation steering the preferred attack method here?
2. Activation steering has been around for quite a while (Turner et al., 2023; Rimsky et al., 2024), and ASA feels like a fairly straightforward application to adversarial settings. What's the core novelty beyond "let's use activation steering for attacks instead of helpful steering"? Is the main contribution just the empirical finding that this works?
3. Given the incremental nature of the attack method, I'd expect the defense results to be quite strong to justify publication. But LAPT's improvements in Table 3 seem modest - often reducing PASR by 0.1-0.2, and sometimes requiring interpolation weights as high as 0.5 (Table 10), which suggests you're barely changing the model. Why should I be convinced that LAPT is a meaningful contribution rather than just "train on adversarial examples with some layer-specific targeting"?
4. You use "safety alignment" throughout, which isn't standard terminology in the field - most work just says "alignment" or "safety training." Is there a specific reason for this choice, or is it just a writing quirk? More broadly, the paper has quite a few awkward phrasings and what appear to be grammatical errors (verb conjugation issues, etc.). Have you had a native English speaker review this?
5. You claim existing methods focus on "surface-level refusal behaviors" rather than "internal representations" (page 1, abstract). But what does this actually mean? RLHF and DPO do shape internal representations through training - they don't just pattern-match refusals. Can you be more precise about what makes current alignment "shallow" and what would constitute "deep" alignment?
6. You've put significant effort into ASABench (4,862 samples), but I'm not sure who the target audience is. If the threat model isn't realistic, why do we need a benchmark for evaluating defenses against it? What would change in the field if ASABench became widely adopted?
7. Have you compared LAPT to just training on more diverse refusal examples or using existing robustness techniques? The "layer-wise" aspect seems like it could just be added complexity - have you ablated this to show it matters beyond standard adversarial training?

---

### Official Review · Reviewer_mKzK · 2025-11-02

**Soundness:** 2
**Presentation:** 2
**Contribution:** 2
**Rating:** 2
**Confidence:** 4

**Summary:**

The paper introduces a steering attack method that maximizes the change in a model's output from a perturbation to the activations. They include a targeted attack that tries to generate a compliant response (i.e., one that starts with a prefix that indicates compliance with a harmful request) and maximize the likelihood of that response. They show that small perturbations can break safety alignment for models. They design an algorithm to counteract this that introduces perturbations into model activations and trains the output to have the original logit values. They show that this reduces the attack success rate of their attack.

# Contributions

* They analyze the robustness of model outputs to latent perturbations
* They introduce a training method to reduce this vulnerability

**Strengths:**

Overall, the paper studies an important problem and the systematic approach to measuring the robustness of models to perturbations of their internal representations appears to be a novel contribution. Their training methodology seems potentially useful as it uses different objectives from standard latent adversarial training. Overall, the paper seems like it may describe a novel latent attack methodology and the particular method of using a target compliant suffix to generate the attacks seems novel and potentially valuable.

**Weaknesses:**

The paper does not appropriately characterize itself with respect to the related literature and suffers from clarity issues about its contributions or evaluations. Within the main body of the paper, related work is discussed in a single paragraph and primarily provides a generic overview. Discussion of specific related work is delegated to an appendix. This makes it quite hard to evaluate the paper and creates a misleading perception of the work.

A particular concern is that the paper does not appropriately situate itself with respect to the existing literature on latent-space attacks for models. The primary example of this is Latent Adversarial Training, which is cited in the appendix, but it is unclear how or why this approach is different or better. For example, how does the $ASA_{grad}$ attack differ from the latent attacks used in LAT? The distinction is unclear, and there is no comparison between the methods.

Beyond this, in the experiments section, it is hard to understand exactly what is being evaluated. For example, in the evaluation og LAPT, it is unclear whether the attack success rates are for random perturbations or for the more targeted attack. This seems quite important for interpreting the results. In general, there is not enough information in the body of the paper that I would be confident in the ability of a member of the community to replicate the benchmark provided.

**Questions:**

* Can you clarify how your attack methodology differs from the attacks considered in Latent Adversarial Training?
 * Are your evaluations against random or adversarial attacks in the results section?
 * How do you plan to restructure the paper in order to more clearly discuss related work and identify the novelty of the method?
 * Can you clarify how the 'fine-grained layer-wise vulnerability analysis' of ASABench actually works? It's not clear to me what this means. For example, what is a validated attack instance in this case? Is it a combination of a model and an input?

---

### Note · Authors · 2026-01-05

I have read and agree with the venue's withdrawal policy on behalf of myself and my co-authors.